# Analysis of Nanofluids Behavior in a PV-Thermal-Driven Organic Rankine Cycle with Cooling Capability

**Samuel Sami** [1,2]

1   Research Center for Renewable Energy, Catholic University of Cuenca, Cuenca 010107, Ecuador; dr.ssami@transpacenergy.com
2   TransPacific Energy, Inc., Las Vegas, NV 89183, USA

**Abstract:** This paper discusses the performance of nanofluids in a PV Thermal-driven Organic Rankine Cycle (ORC) with cooling capabilities. This study was intended to investigate the enhancement effect and characteristics of nanofluids; $Al_2O_3$, $CuO$, $Fe_3O_4$ and $SiO_2$ on the performance the hybrid system composed of PV Thermal, ORC and cooling coil. The quaternary refrigerant mixture used in the ORC cycle to enhance the ORC efficiency is an environmentally sound refrigerant mixture composed of R152a, R245fa, R125, and R1234fy. It was shown that the enhancement of the efficiency of the hybrid system in question is significantly dependent upon not only the solar radiation but also the nanofluids concentration and the type of nanofluid as well as the fluid temperature driving the ORC. A higher hybrid system efficiency has been overserved with nanofluid CuO. Moreover, it has been also shown that on the average, the hybrid system efficiency was higher 17% with nanofluid CuO compared to water as the heat transfer fluid. In addition, it was also observed that the higher cooling effect produced is significantly increased with the use of the nanofluid CuO compared to the other nanofluids under investigation and water as heat transfer fluid. The results observed in this paper on ORC efficiency and PV solar panel efficiency are comparable to what has been published in the literature.

**Keywords:** PV-thermal solar collector; nanofluids; organic rankine cycle; modelling; simulation

## 1. Introduction

Solar energy is regarded as a favorable and clean renewable energy resource. A PV-Thermal solar collector, as the heat source of an Organic Rankine cycle (ORC) can be used for power generation. Different types of different nanofluids can be considered for use in the solar PV-Thermal to improve its efficiency. In the following sections, we will discuss the literature reported work [1–28] on Organic Rankine Cycle, ORC, solar PV-Thermal, and different nanofluids as a heat transfer fluid to drive waste heat boiler of ORC.

As solar irradiance is converted into electrical energy in the PV cell, excess thermal energy is generated due to the inherent conversion efficiency process limitation of the PV cell. The higher the excess thermal, the higher the temperature of the PV cell. This, in turn, reduces the conversion efficiency of the cell. By using a flow of cold-water flow though the thermal tubing underneath the PV cell, this excess thermal energy can be recovered for a useful purpose. This process reduces the PV cell temperature and enhances the energy conversion efficiency of the PV solar panels [1–27].

Nowadays, the organic rankine cycle (ORC) has become an important technology for recovering waste heat, heat from renewable heat source for improving thermal performances. This technology is manly used for low temperature and low-grade energy resource [6,7,18]. ORC can be integrated with

different working thermal power plants such as a combined gas-steam power plant, a solar-integrated combined cycle, a solid oxide fuel cell, geothermal, biomass or combination of these hybrid power plants to be used as a heat source to ORC.

A paper has been presented in [9] on a theoretical investigation of a new configuration of the combined power and cooling cycle known as the Goswami cycle ammonia as the refrigerant. A comprehensive analysis was conducted to determine the effect of key operation parameters such as ammonia mass fraction at the absorber outlet and boiler-rectifier on the power output, cooling capacity, effective first efficiency, and effective exergy efficiency. Moreover, Guzmán et al. [9] showed that the new dual-pressure configuration generated more power than the single pressure cycle, However, the results also showed that it reduced the cooling output as there was less mass flow rate in the refrigeration unit.

A review of three cycles, namely the Organic Rankine cycle, the Kalina cycle and the Goswami cycle and various work on them was presented by Karimi et al. [10], work done by different authors on optimization of the cycles and introduction of new efficient cycles. Selection of fluids, three cycles namely the Kalina cycle, the Goswami cycle and ORC, have been optimized by different authors using different types of fluids and their reviews are concluded here. Low-grade heat topic deals with the electricity generation from low-grade heat sources such as solar thermal, geothermal and industrial waste heat using different cycles mainly ORC, the Goswami cycle and the Kalina cycle. In addition, the authors presented an analysis of different thermodynamic cycles for combined power plant using low-grade heat sources. Different thermodynamic cycle using low grade heat sources for combined power plant were also reviewed.

A theoretical investigation of a combined Power and Cooling Cycle that uses an Ammonia-Water mixture has been reported by Demirkaya et al. [11]. The cycle combines a Rankine and an absorption refrigeration cycle. A comprehensive analysis of the effect of several operation and configuration parameters, including the number of turbine stages and different superheating configurations, on the power output and the thermal and exergy efficiencies was conducted. Their results show that the Goswami cycle can operate at an effective exergy efficiency of 60%–80% with thermal efficiencies between 25% and 31%. This study demonstrated that for multiple turbine stages, the use of partial superheating with Single or Double Reheat stream showed a better performance in terms of efficiency. It also showed an increase in exergy destruction when heat source temperature was increased.

A combined thermal power and cooling cycle which combines the Rankine and absorption refrigeration cycles has been presented by [12,13]. Ammonia-water mixture was used as a working fluid. This cycle is ideally designed for solar thermal power using low-concentrating solar collectors. This cycle can also be used as a bottoming cycle for any thermal power plant.

Karaal [14] presented an exergetic analysis of a combined power and cooling cycle that uses ammonia-water mixture as the working fluid [15]. He reported that such cycles use solar or geothermal energy or waste heat energy from a conventional power cycle. He also pointed out that an ammonia-water power cycle can be used as an independent cycle to provide the power output and cooling. Karaal reported that the boiling process and a heat transfer process at low temperature destruct the energy given to the boiler so that the energy efficiency is low; however, the exergy efficiency was higher than the energy efficiency. It was also reported by Karaal, increasing the turbine inlet pressure decreases the energy and exergy efficiencies.

Currently, interest is increasing in hybrid photovoltaic/thermal systems, which are a continuation of the photovoltaic solar system [16–28]. The PV solar panels and the thermal systems are integrated into one system known as photovoltaic (PV/T). The conversion efficiency of solar radiation to electricity in the solar cell ranges from 6% to 15% [24]. The rest of the fallen radiation is reflected, and the other part dissipated and absorbed as heat. Since PV cells must be placed under direct sunlight to produce electricity, their high temperature is inevitable [24]. Therefore, the PV/T systems of today are being extensively studied; in these systems, a heat exchanger is added to the PV cell to extract the excess heat, increase the electrical efficiency of the cell and utilize the heat absorbed by the PV cells

in other applications [24–26]. The heat exchanger used here is either an aluminum rectangular pipe or a tank set mounted on the back of the PV panels and through which the refrigerant fluids flow. Sami et al. [24–26] developed and presented several numerical models to predict the performance of PV-Th under different conditions such as solar radiations from 500 to 1200 W/m2, heat transfer fluid flow in the back-heat exchanger, different nanofluids as heat transfer fluids and various cell temperatures [24–26].

Many researchers have conducted numerical and experimental studies to improve thermal conductivity using nanofluids and in various mixing methods. Reference [28] declared that thermal conductivity increased with temperature when suspended with nanostructure $Al_2O_3$, CuO and water was used. The results of the study showed that the cooling fluid, which contains smaller particulate nanoparticles (CuO), show a thermal conductivity greater than the larger particles. Reference [28] developed two experimental equations to evaluate the effective thermal conductivity o nanomaterials and dynamic viscosity. In this study, the improvement in heat transfer by the dispersion of solid nanoparticles in the base fluid was calculated with different weights and temperatures ranging from 21 °C to 51 °C. The researchers used nanoparticles Cu, $Al_2O_3$ and $TiO_2$ with cooling fluid (water and ethylene glycol). Al-Waeli et al. [27] initiated a numerical simulation of heat transfer in a wavy channel and the improvement obtained using nanofluids made of nano-copper and water. The results showed that the use of nano cooling fluid with wavy walls enhanced the heat exchange between the wall and the flow significantly. The results of the study showed that the addition of nanoparticles by 10% to water resulted in improved heat exchange by 25%. Furthermore, A.H.A. Al-Waeli et al. [27] concluded that among these seven mechanisms, thermophoresis and Brownian diffusion can be considered the most important. The study also showed clearly that nanoparticles move homogeneously with fluid in the presence of turbulent eddies, so their negative impact on the density of disturbance is doubtful.

The organic rankine cycle (ORC) has been presented in the literature as a combined power cycle, without any cooling capability except the concept presented by [13], where ammonia was used as the refrigerant, with a low temperature heat source. However, this study presents a new concept of a hybrid system using an organic rankine cycle ORC using refrigerant mixture powered by PV-Thermal solar panels and integrated/built in a cooling coil. This new concept presented hereby is intended to highlight and discuss the use of PV-Thermal solar panel to produce power by an ORC and supply cooling capabilities driven by solar energy. This research work focuses in particular on the performance inherent parameters of the new concept system such as the hybrid system efficiency and the cooling effect produced. The study implements a numerical finite difference model based on conservation equations to predict the inherent parameters of the system and their impact on the system performance.

## 2. Mathematical Model

The mathematical model presented hereby was established based on the mass and energy equations written to describe the behavior the nanofluids circulating in the PV thermal solar collectors, driving an Organic Rankine Cycle, ORC, with a quaternary refrigerant mixture as shown in Figure 1. The solar radiation was absorbed by the PV solar panels and converted into electricity and thermal energy. The latter is dissipated and heats up the nanofluids heat transfer fluid that is used to drive the waste heat boiler of the ORC and generate vapor refrigerant. In the turbine, the thermal energy is converted into kinetic energy and produces power at the turbine shaft and the generator. The low-pressure vapor exiting the turbine is condensed in the condenser into liquid and pumped back to the waste heat boiler through the cooling/freezing coil and the regenerator, as illustrated in Figure 1. The refrigerant mixture circulates in the ORC loop is an environmentally sound quaternary mixture and composed of R512a, R125, R1234fy, and R245fa with a boiling temperature of −28.13 °F, a critical temperature of 220.67 °F at a critical pressure of 59.85 Psi. The thermodynamic and thermophysical properties of the refrigerant mixture were obtained at REFPROP [15]. REFPROP is a computer program distributed through the Standard Reference Data program of NIST that provides the thermophysical properties of pure fluids and mixtures over a wide range of fluid conditions, including liquid, gas, and

supercritical phases. It contains critically evaluated mathematical models with the goal of representing the properties to within the uncertainty of the underlying experimental data used in their development.

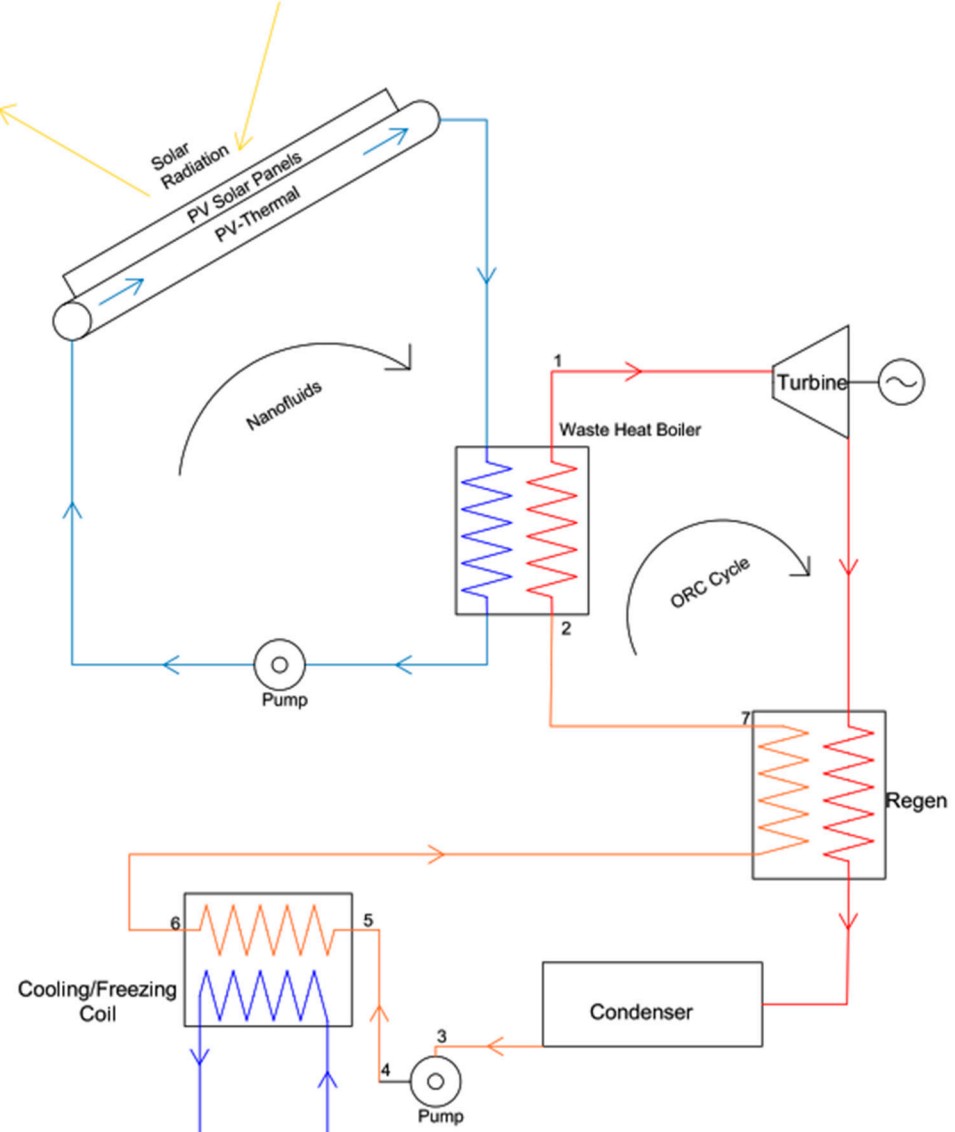

**Figure 1.** Organic rankine cycle with cooling capability.

In the following sections, the different equations of mass and energy are written and presented for each definite control volume element of the nanofluids loop and ORC cycle presented. It is assumed in the model that the nanofluid is homogeneous, isotropic, incompressible, and Newtonian, that inlet velocity and inlet temperature are constant, and that the thermophysical properties of the nanofluids are constant.

## 2.1. PV Thermal Model

The following thermal analysis was performed for the PV cell; however, it is assumed that all PV cells behaved the same; therefore, it was applied to the PV solar panel. The heat absorbed by the PV solar cell can be calculated by the following [6,24–27]:

$$Q_{in} = \alpha_{abs} G S_p \tag{1}$$

where $\alpha_{abs}$: overall absorption coefficient, $G$: total Solar radiation incident on the PV module and $S_p$: total area of the PV module.

Meanwhile, the PV cell Temperature is computed from the following heat balance:

$$mC_{p\_module}\frac{dT_C}{dt} = Q_{in} - Q_{conv} - Q_{elect} \tag{2}$$

where

$T_C$: PV cell temperature
$mC_{p\_module}$: thermal capacity of the PV module
$t$: time
$Q_{in}$: energy received due to solar irradiation
$Q_{conv}$: energy loss due to convection
$Q_{elect}$: electrical power generated

and the Solar energy absorbed by the PV cell, Qin, is given by Equation (1).

### 2.2. PV Model

The solar photovoltaic panel is constructed of various modules and each module consists of arrays and cells. The dynamic current output can be obtained as follows [6,24–27]:

$$I_P = I_L - I_o\left[\exp\left(\frac{q(V + I_P R_S)}{AkT_C} - \frac{V + I_P R_S}{R_{sh}}\right)\right] \tag{3}$$

$I_p$: output current of the PV module
$I_L$: light generated current per module
$Io$: reverse saturation current per module
$V$: terminal voltage per module
$R_s$: diode series resistance per module
$R_{sh}$: diode shunt resistance per module
$q$: electric charge
$k$: Boltzmann constant
$A$: diode ideality factor for the module.

The AC power is calculated using the inverter efficiency $\eta_{inv}$, output voltage between phases, neutral for single-phase current $I_o$ and $cos\varphi$ as follows [6,24–27]:

$$P(t) = \sqrt{3}\,\eta_{inv}V_{fn}I_o\,cos\varphi \tag{4}$$

### 2.3. ORC Model

The energy balance at the ORC cycle gives the following [6,7,10]:

$$W_{ORC} = m_{ref}(h_1 - h_2) \tag{5}$$

$$Q_{WHB} = m_{ref}(h_1 - h_4) \tag{6}$$

$$Q_{COND} = m_{ref}(h_2 - h_3) \tag{7}$$

$$W_{PORC} = m_{ref}(h_4 - h_3) \tag{8}$$

$$Q_{Cc} = m_{ref}(h_6 - h_5) \tag{9}$$

$$Q_{regcn} = m_{ref}(h_7 - h_6) \tag{10}$$

where $h$ is the enthalpy

$h_1$: enthalpy at the outlet of the waste heat boiler (kj/Kg)
$h_2$: enthalpy at the exit of the vapor turbine (kj/Kg)
$h_3$: enthalpy at the condenser outlet (kj/kg)
$h_4$: enthalpy at ORC pump outlet (kj/kg)
$h_5$: enthalpy at inlet of cooling/freezing coil (kj/kg)
$h_6$: enthalpy at outlet of cooling/freezing coil (kj/kg)
$h_7$: enthalpy at the outlet of regenerator (kj/Kg)
$m_{ref}$: refrigerant mass flow rate (kg/s)

The ORC-PV hybrid system efficiency can be calculated as

$$\eta_{ORC-h} = \frac{W_{ORC} + p(t) + Qcc - W_{P_{ORC}}}{Q_{in}} \qquad (11)$$

where

$W_{ORC}$: power produced by ORC (KW)
$p(t)$: PV solar output (kW) defined by Equation (4)
$Qcc$: cooling coil thermal capacity (kw) and defined by Equation (9)
$W_{P_{ORC}}$: pump power consumption defined by Equation (8)
$Q_{in}$: solar radiation (kw) and defined by Equation (1)

## 3. Nanofluid Heat Transfer Fluid

Sharama et al. [17] and Sami [18] presented equations to calculate the thermophysical and thermodynamic properties of nanofluids such as specific heat, thermal conductivity, viscosity and density, using the law of mixtures in terms of the volumetric concentration of nanoparticles;

$$\alpha_{total} = \alpha_{particles} + \alpha_{base\ fluid} \qquad (12)$$

where $\alpha$ represents a particular thermophysical property of the nanofluid under investigation.
The nanofluid thermal and thermophysical properties, $\alpha_{total}$, can be calculated as follows [19,20]:

$$\alpha_{total} = \alpha_{base\ fluid} + \alpha_{particles}(\Phi) \qquad (13)$$

where $\Phi$ represents the nanoparticles volumetric concentration.
The thermal conductivity is related to thermal diffusivity and density of the nanofluids as follows:

$$\lambda = \alpha\ \delta\ C_p \qquad (14)$$

where $Cp$ is the specific heat, $\alpha$ is the thermal diffusivity, $\lambda$ and $\rho$ represent the thermal conductivity and density, respectively.
The specific heat is calculated for nanofluids as follows [11–13]:

$$c_{pnf} = \frac{(1-\varnothing)(\rho Cp)_{bf} + \varnothing(\rho Cp)_p}{(1-\varnothing)\rho_{bf} + \varnothing\rho_p} \qquad (15)$$

where "*nf*" and "*bf*" refer to the nanofluid and the basic fluid, respectively. $\varnothing$ is the nanofluid particle concentration. $\rho$ represents the density.

The density of nanofluids can be written as follows [5–7,17–20]:

$$\rho_{nf} = \varnothing_p \rho_p + (1 - \varnothing)\rho_{pf} \tag{16}$$

where

$\rho_{pf}$: density of the nanoparticle.

## 4. Numerical Procedure

The energy conversion process in the PV-Th panels, integrated ORC and cooling described by Equations (1)–(16) and presented in Figure 1 was programmed and solved according to the logical diagram presented in Figure 2. The calculation starts with the input of the parameters of the PV-Thermal solar panel, thermal tubes, and characteristics of the nano particles; $Al_2O_3$, CuO, $Fe_3O_4$ and $SiO_2$ and water as the heat transfer fluid. The system equations were integrated into the finite-difference formulations to determine the behavior of the process shown in Figure 1. Iterations were performed using MATLAB iteration techniques until a converged solution was reached with less than 0.05. With the known values of the solar radiation, the mass flow rate of the nanofluid circulating in the thin tubes welded to the PV solar collector was determined. Then, the thermophysical properties and the heat transfer characteristics of the base fluid, water, and nanofluids at different concentrations were determined. Then, the parameters describing the behavior of PV-Thermal solar panels, ORC and the cooling coil were determined at different conditions. Finally, the hybrid system efficiencies were calculated.

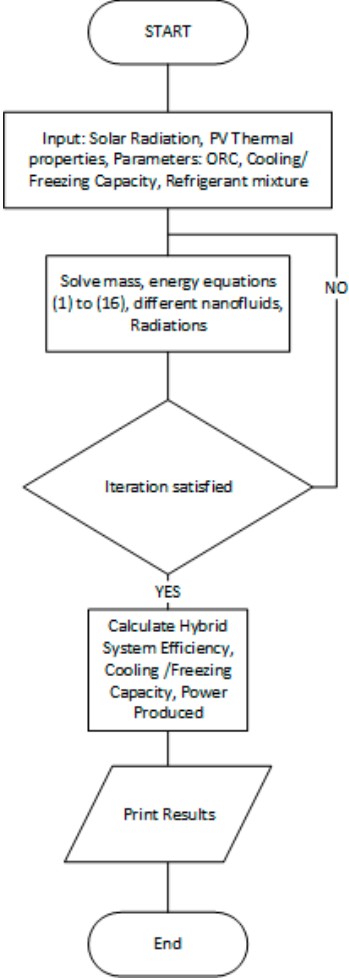

**Figure 2.** Logical diagram for numerical solution.

## 5. Discussion and Analysis

The system of Equations (1)–(16) has been numerically solved in finite-difference formulation for the energy conversion process in the hybrid system of PV-Thermal solar collector, ORC with cooling capability employing nanofluids; $Al_2O_3$, $CuO$, $Fe_3O_4$ and $SiO_2$ at different concentrations and water as base heat transfer fluid. Table S1 presents the thermophysical properties of nanofluids used in this study. In the following sections, the predicted results are presented under different inlet conditions, such as solar insolation, heat transfer fluid flow rates from the PV-Thermal, heat transfer fluid temperatures and various nanofluids at different volumetric concentrations. In the numerical simulation, 100 PV solar panels were assumed with 300 watts per each PV solar panel. Solar radiations were taken as 500, 750, 1000 and finally, 1200 $w/m^2$ and heat transfer fluid temperature varied from 176 °F to 212 °F. As reported in Sami and Marin [7], calculations using Equations (1)–(4) yielded the efficiency of the PV solar panels used in this study varies between 19% and 23% depending on the solar radiation that varies between 500 $w/m^2$ to 1200 $w/m^2$. These values will be taken and considered as base values for comparison between the hybrid system efficiency of the PV-Thermal and ORC system and that of the PV solar panels. The heat transfer fluid flow rate circulating in the loop, driving the waste heat boiler of the ORC between 8.89 GPM (4208 lb/hr) to 22.6 GPM (10550 lb/hr) at temperatures varied between 176 °F to 212 °F. For comparison purposes, the pressure in the refrigerant mixture cycle of the ORC was kept constant between the waste heat boiler and the condenser at 133 psi and 55 psi, respectively. The boiling temperature of the refrigerant mixture was at –28 °F. The refrigerant mixture exits the waste heat boiler to the vapor turbine at vapor saturation conditions. Appropriate heat exchanger and vapor turbine efficiency values were used for the ORC cycle calculations as well as the cooling capacity. The Carnot cycle efficiency of the ORC conditions was 27.38% and on average, the ORC efficiency was 8.3% using the environmentally sound quaternary refrigerant mixture composed of R152a, R245fa, R125, and R1234fy. In the following sections, the impact of the solar radiations on the Hybrid system performance, and cooling/freezing effect produced as defined by Equation (9) and the efficiency of the hybrid system composed of the PV solar panel, PV-Thermal and the ORC, as calculated by Equation (11) are analyzed and discussed. The aim of this study was to illustrate the role of nanofluids in the enhancement of the Hybrid system efficiency and the cooling effect produced over that of the PV solar panels and ORC, respectively.

The hybrid system efficiency calculated by Equation (11) is plotted in Figures 3–6 to illustrate the impact of solar radiations and the different nanofluids and water as the heat transfer fluid on the hybrid system performance of the PV-Thermal solar panels. It is quite evident from the results presented in these figures that the nanofluid CuO has the highest efficiency among the different nanofluids and water. It is also evident from these figures that the higher the nanofluid flow temperature, the higher the hybrid system efficiency. This can be interpreted as pointed out by [18–20] that convective heat transfer performance of suspended nanoparticles outstandingly increases the heat transfer thermal capacity of the base-fluid and the nanofluids have higher heat transfer coefficients than those of the base-fluids for the same Reynolds number.

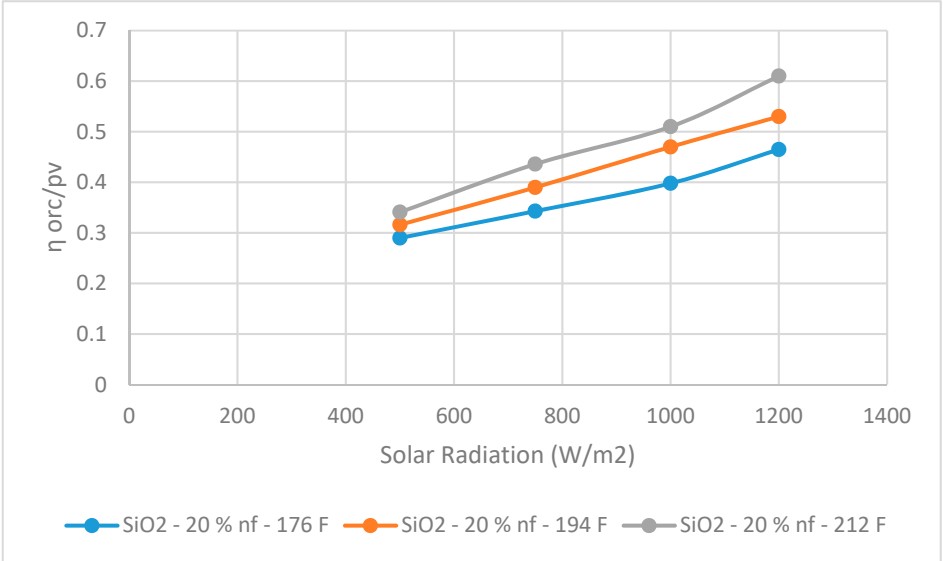

**Figure 3.** Hybrid system efficiency for Nanofluid $SiO_2$ at different temperatures and 20% volumetric concentration.

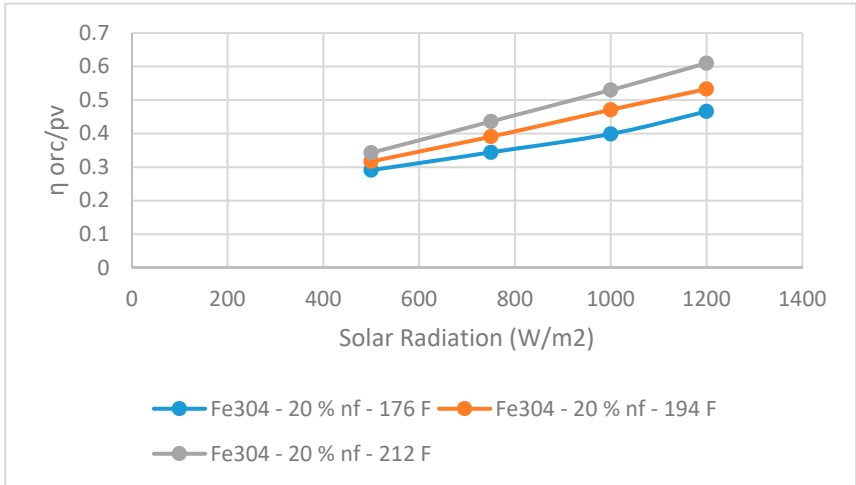

**Figure 4.** Hybrid system efficiency for Nanofluid $Fe_3O_4$ at different temperatures and 20% volumetric concentration.

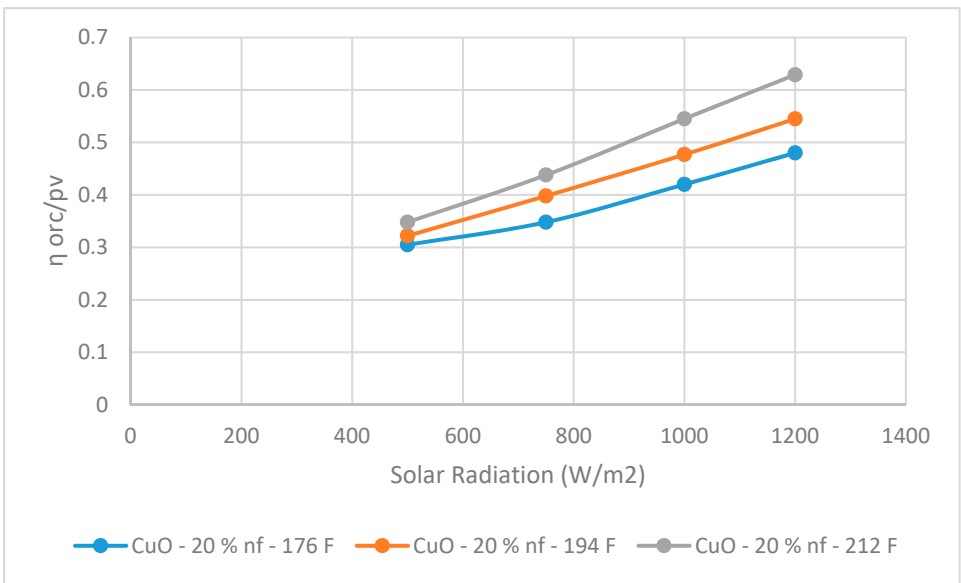

**Figure 5.** Hybrid system efficiency for Nanofluid CuO at different temperatures and 20% volumetric concentration.

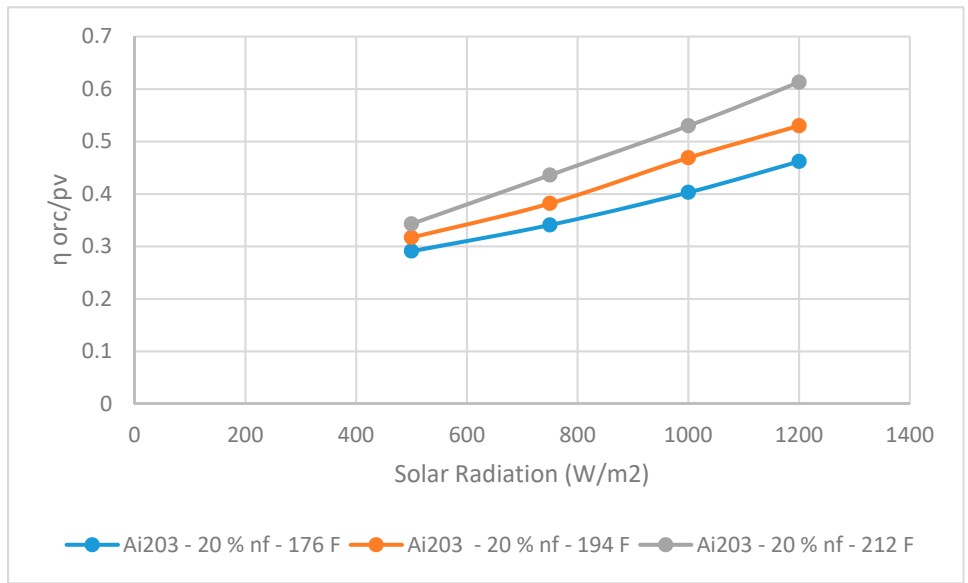

**Figure 6.** Hybrid system efficiency for Nanofluid $Al_2O_3$ at different temperatures and 20% volumetric concentration.

It can also be noted from samples of the results presented in Figures 3–10 as well as previous predictions that such an improvement in the heat transfer capacity becomes more significant with the increase in the volume concentration of the particle loading in nanofluids. This enhancement in the heat transfer capacity of the heat transfer fluid increases the thermal energy delivered to the waste heat boiler of the ORC and thus, increases the power output of the ORC and the ORC efficiency. Therefore, the results presented in the aforementioned figures and others obtained in this study show that the enhancement of the efficiency of the hybrid system in question is significantly dependent upon not only the solar radiation but also on the nanofluids concentration and the type of nanofluid as well as the fluid flow temperature driving the ORC. In addition, elsewhere in the paper, a comparison between the different nanofluids is presented to show the enhancement of the hybrid system efficiency using nanofluids over the water as heat transfer fluid.

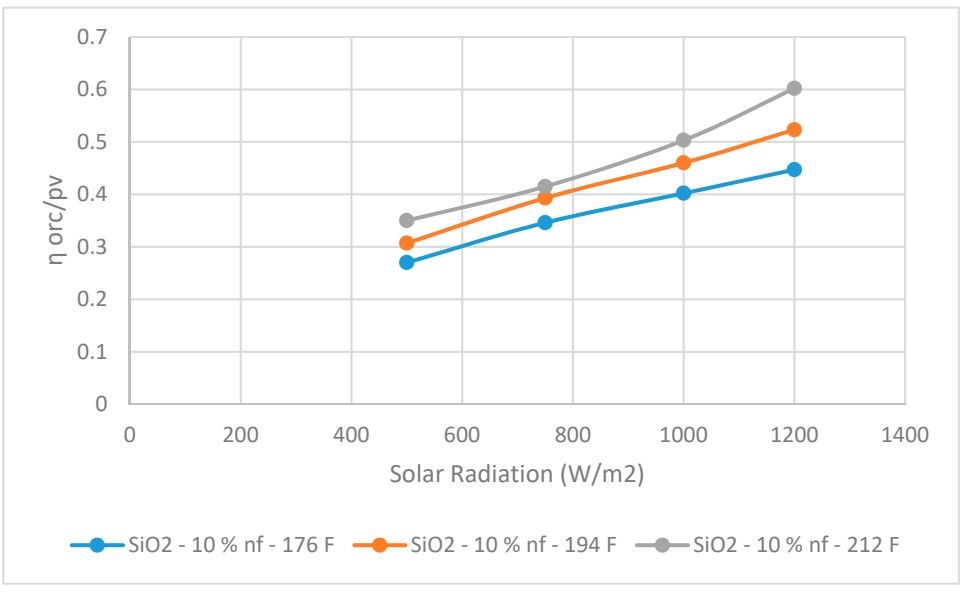

**Figure 7.** Hybrid system efficiency for Nanofluid $SiO_2$ at different temperatures and 10% volumetric concentration.

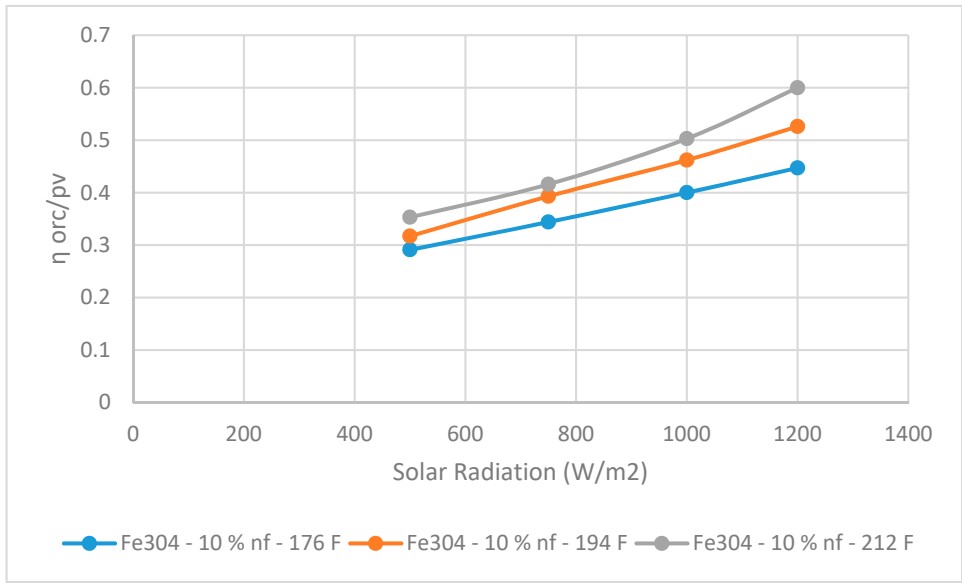

**Figure 8.** Hybrid system efficiency for Nanofluid $Fe_3O_4$ different temperatures and 10% volumetric concentration.

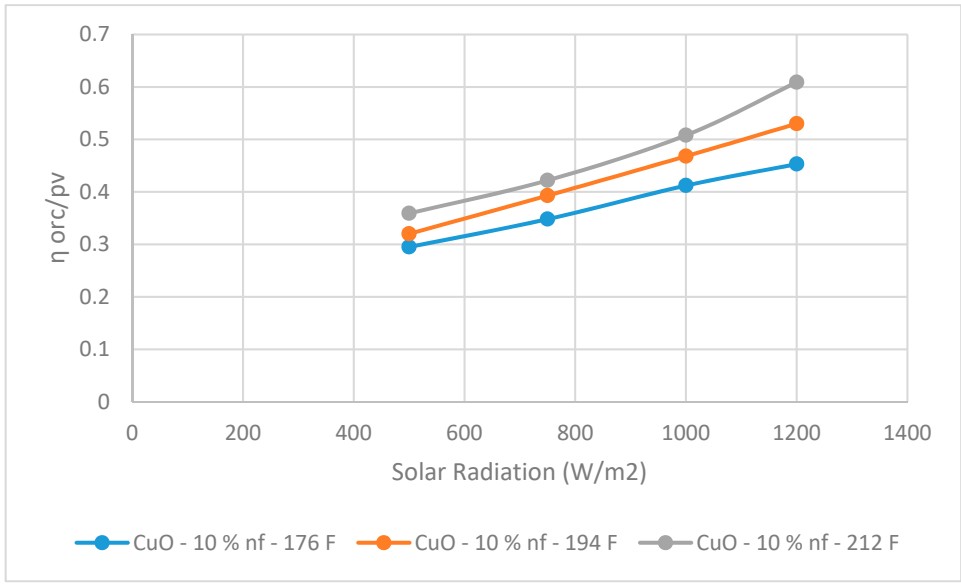

**Figure 9.** Hybrid system efficiency for Nanofluid CuO different temperatures and 10% volumetric concentration.

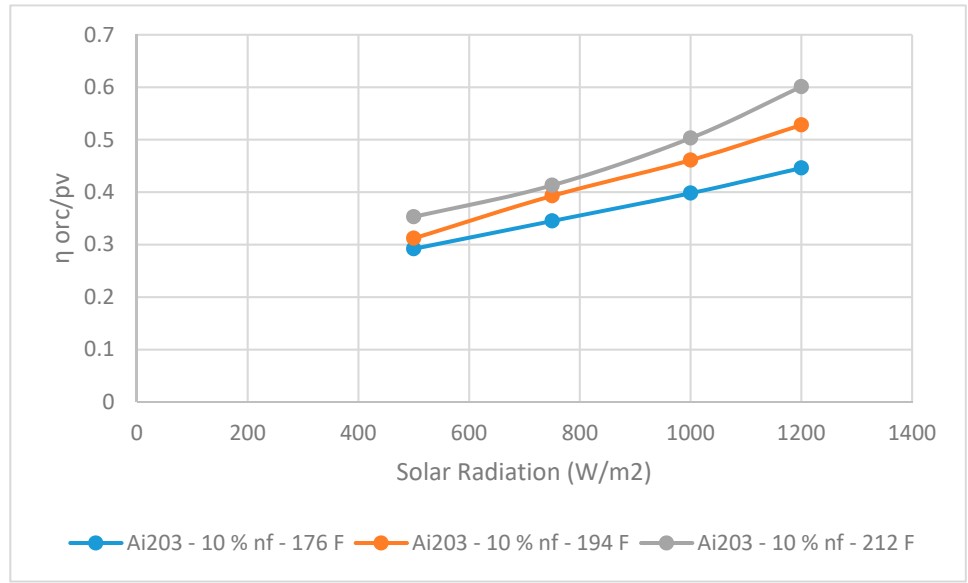

**Figure 10.** Hybrid system efficiency for Nanofluid $Al_2O_3$ different temperatures and 10% volumetric concentration.

Moreover, the results presented in the aforementioned figures as well as others obtained also demonstrate that a higher hybrid system efficiency has been oberserved with nanofluid CuO than the other nanofluids under investigation and water as the heat transfer fluid. This has been reported in previous studies, namely Sami [18], Mohammad et al. [19] and Lazarus et al. [20]. The transport properties, such as density, specific heat, heat capacity, viscosity, and thermal conductivity of nanofluids are the main properties that impact the convective heat transfer of nanofluids [18]. It should be noted that these transport properties have unique the function of temperature of the base heat transfer fluid integrated nanofluid. Therefore, as reported in other references, namely [18–23], nanofluid CuO thermo-physical properties are the main driver for the higher hybrid system efficiency when CuO is used as the heat transfer fluid.

The new concept of the ORC cycle presented hereby includes an added feature which is the integration of a cooling capability to the ORC cycle for the cooling or freezing effect, depending upon the boiling temperature of the refrigerant mixture circulating in the ORC cycle. Figures 11–15 show the different parameters that impact the cooling capabilities of this cycle. The results depicted in these figures demonstrate that the cooling effect produced depends upon various parameters, such as the solar radiation, the temperature of the heat transfer fluid and the types of nanofluids and their concentrations. It is quite evident from the results presented in these figures and others obtained during this study with the different nanofluid concentrations that the higher the heat transfer fluid temperature, the higher the cooling effect produced. As discussed elsewhere in this paper in the case of the hybrid system efficiency, it is believed that the enhancement of the cooling effect is due to the higher thermo-physical properties of the nanofluids. In particular, it can be observed from Figure 16 that the nanofluid CuO has the highest cooling effect produced compared to the other nanofluids under investigation including water as heat transfer fluid. This observation is attributed to the fact that the CuO higher thermo-physical properties, which are the main driver for the higher cooling effect produced and consequently, the enhancement of the hybrid system efficiency with the CuO.

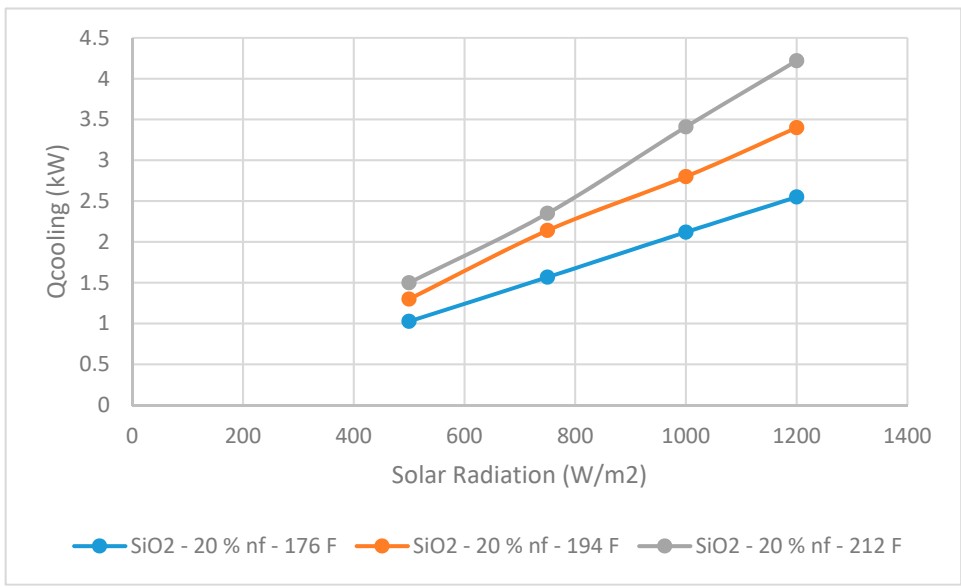

**Figure 11.** Cooling load effect produced with $SiO_2$ at different temperatures.

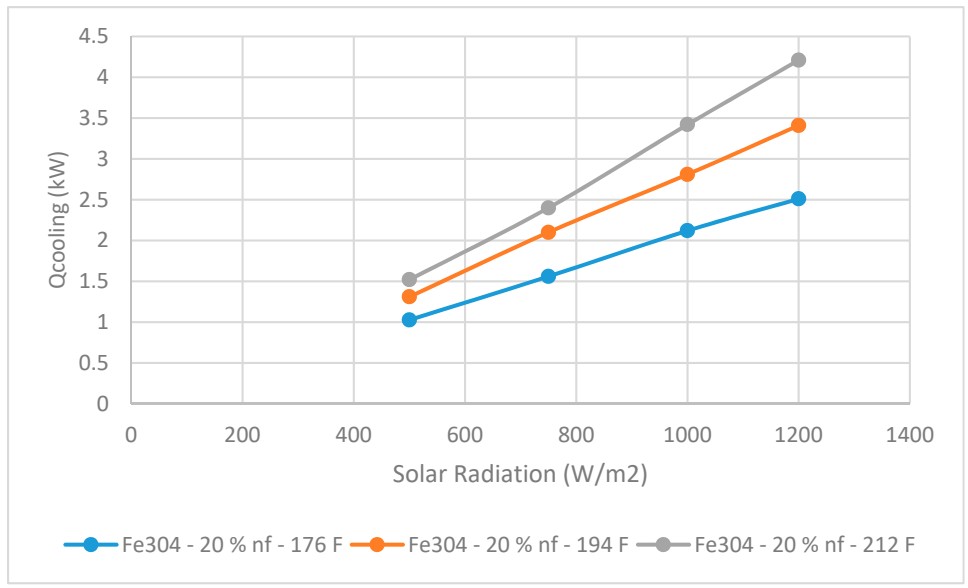

**Figure 12.** Cooling load effect produced with $Fe_3O_4$ at different temperatures.

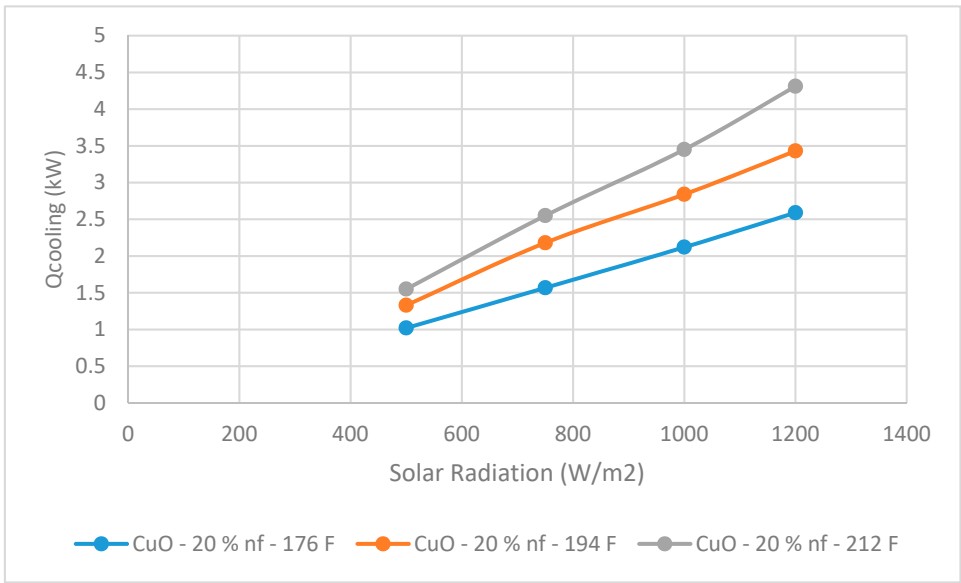

**Figure 13.** Cooling load effect produced with CuO at different temperatures.

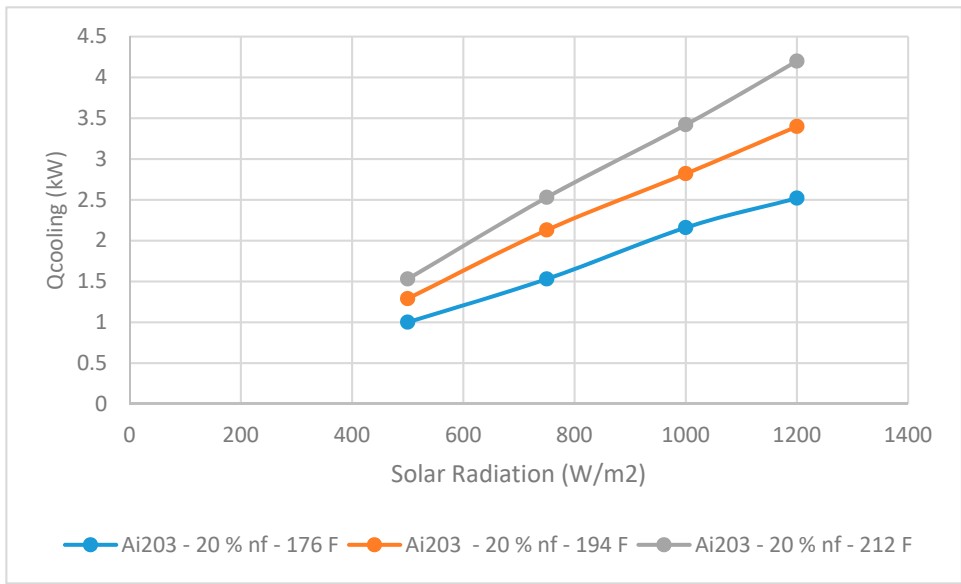

**Figure 14.** Cooling load effect produced with $AI_2O_3$ at different temperatures.

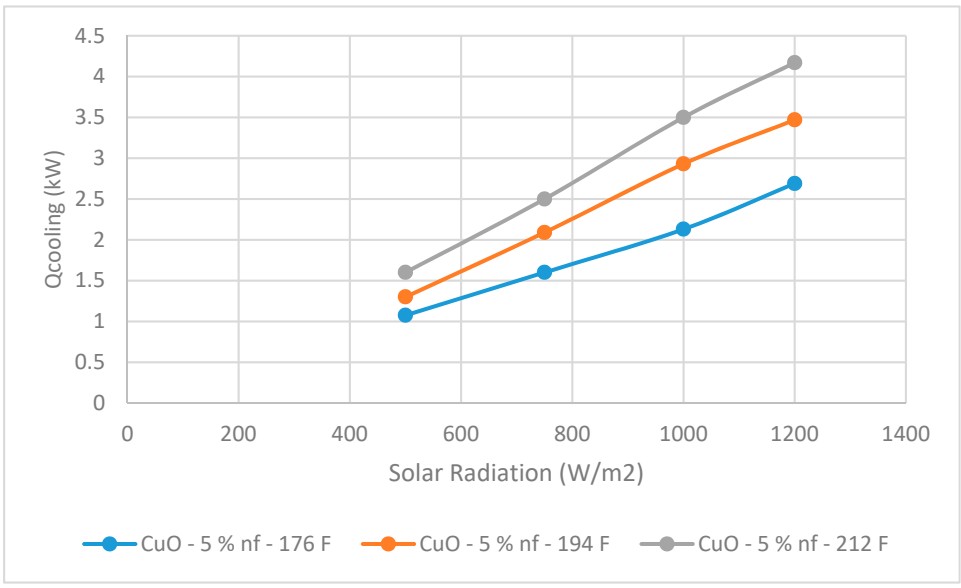

**Figure 15.** Cooling load effect for CuO at 5% concentration.

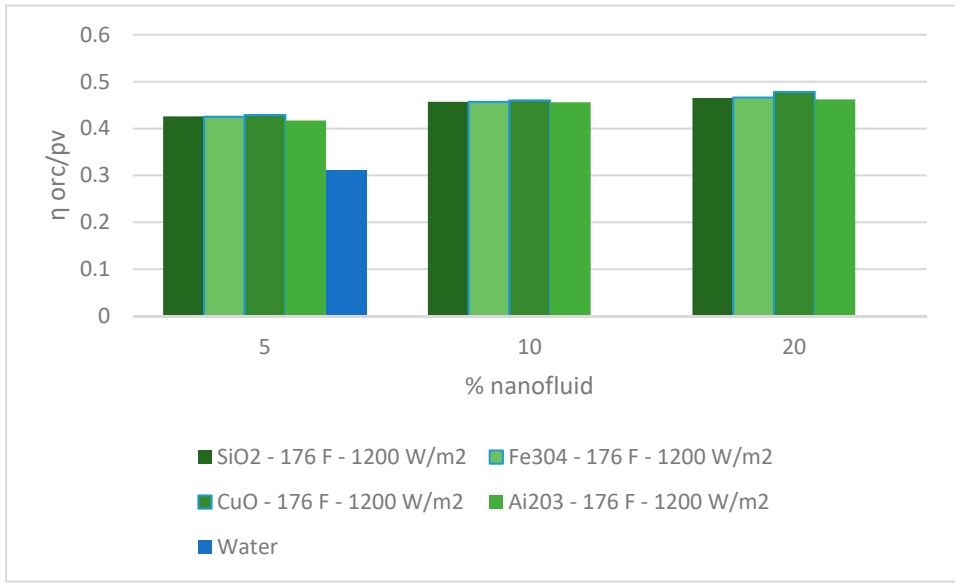

**Figure 16.** Hybrid system efficiency for different nanofluids at 176 °F.

In particular, Figure 16 clearly illustrates that the higher the nanofluid concentration, the higher the cooling effect produced. It has been reported by Sami [18], Mohammad et al. [19] and Lazarus et al. [20] that transport properties such as the density, specific heat, heat capacity, viscosity, and thermal conductivity of nanofluids are function of the nanofluids concentrations and significantly impact the convective heat transfer of nanofluids [18] and obviously the cooling effect produced presented in the aforementioned figures.

A comparison between the different nanofluids under investigation is presented in Figures 16–18 under different heat transfer fluid temperaturs and volume concentrations: 5%, 10% and 20%. To illustrate the effect of different nanolfuids on the efficiency of the hybrid system composed of PV-Thermal and ORC. The depicted results as well as others obtained showed that the nanolfuid CuO as heat transfer fluid experiences the highest hybrid system efficiency compared to the other nanofluids and water as the heat transfer fluid. These findings have been reported by other references, as discussed elsewhere in the paper, namely Sami [18,28]. Clearly, it was also shown that the higher the temperature of the heat transfer fluid, the higher the efficiency of the hybrid system. This is an important finding since just by selecting the right nanofluid for a specific ORC design, the hybrid system efficiency can be significantly enhanced, irrelevent of the solar radiation, heat transfer fluid temperature and the nanofluid concentration. Obviously, as can be illustrated in these figures, the higher heat transfer fluid temperature and the higher the nanofluid concentration the higher the hybrid system effciency.

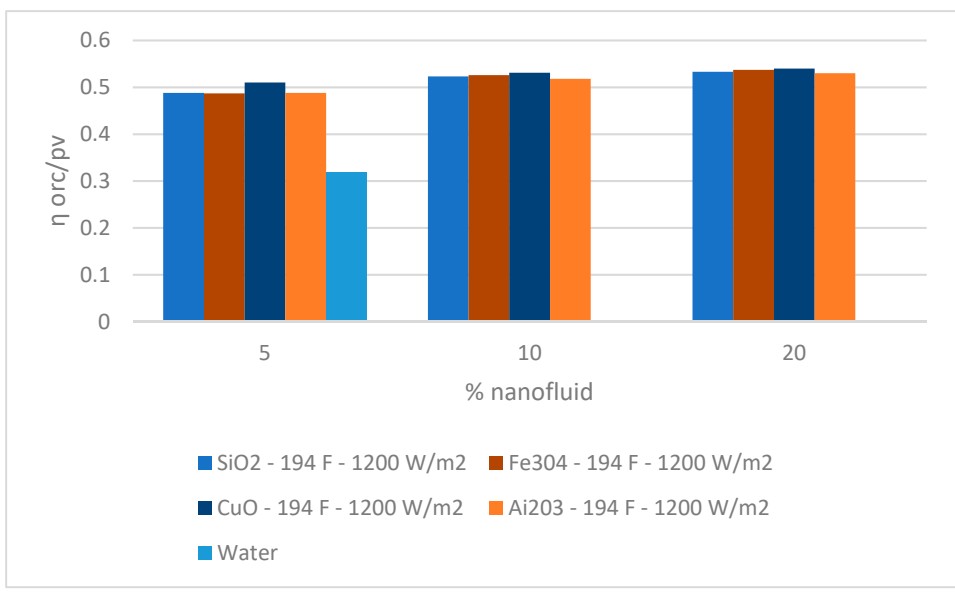

**Figure 17.** Hybrid system efficiency for different nanofluids at 194 °F.

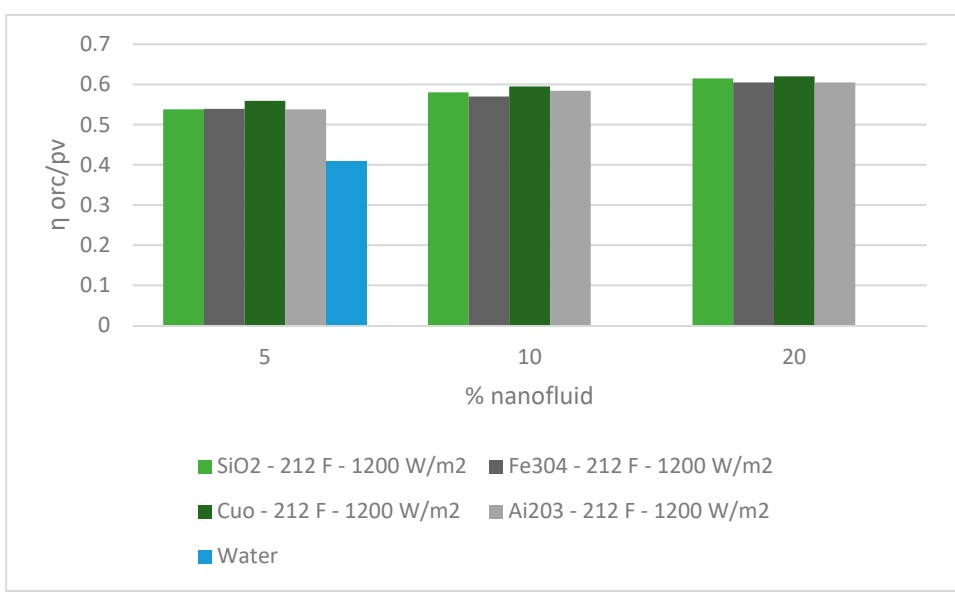

**Figure 18.** Hybrid system efficiency for different nanofluids at 212 °F.

Moreover, as discussed above, the cooling effect produced by this hybrid system is an important feature of the proposed concept since ORC systems are inteded only to produce power. Moreover, as illustrated in this paper, the cooling capabilities are impacted by the different parameters of the PV-Thermal and ORC hybrid system; therefore, Figures 19–21 were created to demonstrate the impact of the different nanofluids and their conditions, heat transfer fluid temperatures and nanofluid volumetric concentrations, as well as solar radiations, on the cooling effect produced by this new hybrid system concept. It is quite clear from the results presented in these figures and others results obtained that the higher the solar radiation and the higher the heat transfer fluid temperatures, the higher the cooling effect produced. However, the data presented in these figures also show that the higher the nanofluid concentrations and the higher heat transfer fluid temperature, the higher the cooling effect produced. Moreover, it was also observed from the results obtained that a higher cooling effect can be produced with the use of the nanofluid CuO compared to the other nanofluids under investigation and water as the heat transfer fluid. This finding is consistent with what has been reported in the literature. It is

belived that this is attributed to the fact that the nanolfuid CuO has higher thermo-physical properties compared to the other nanolfuids, including water as the heat transfer fluid.

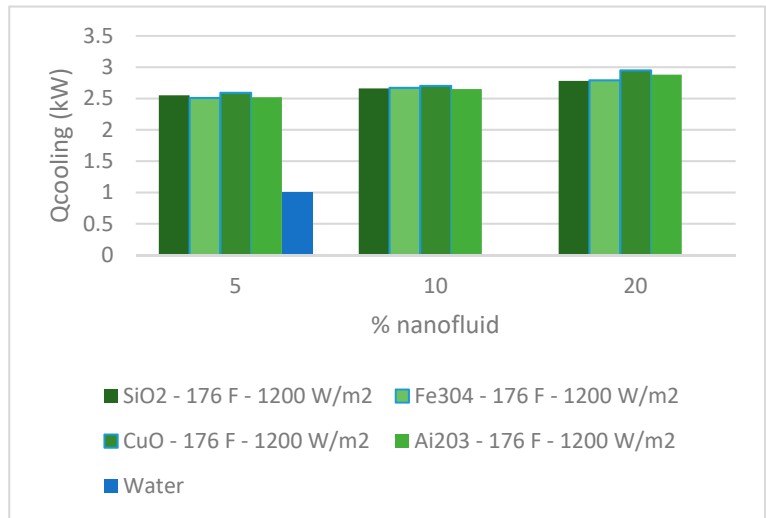

**Figure 19.** Hybrid system cooling effect for different nanofluids.

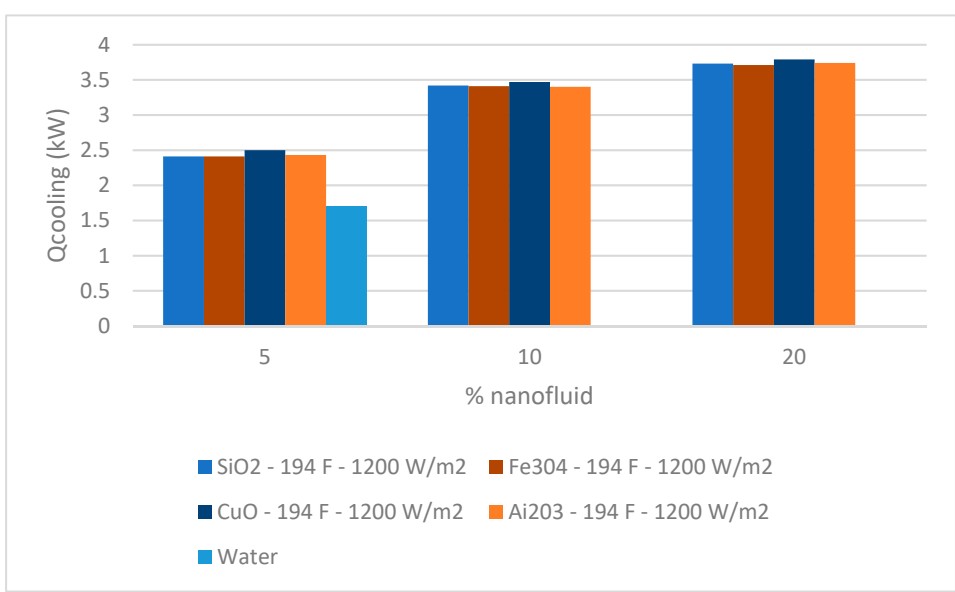

**Figure 20.** Hybrid system cooling effect for different nanofluids.

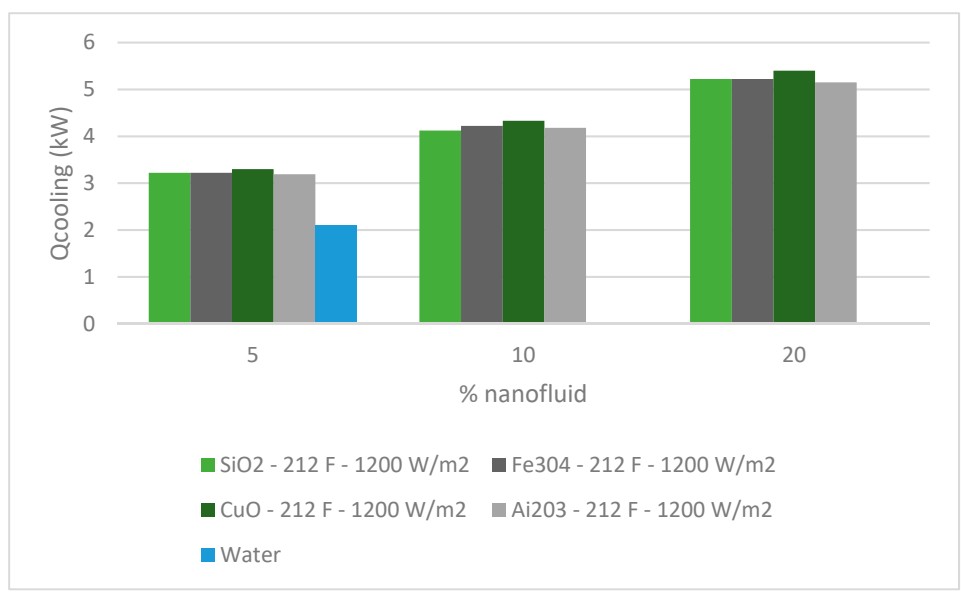

**Figure 21.** Hybrid system cooling effect for different nanofluids.

Due to the lack of experimental data published in the literature on the new concept presented hereby, to validate our numerical model, the predicted values of the efficiency of the ORC cycle at the specific conditions reported in this study were compared with what has been published in the literature on low-temperature applications of ORC, namely Sami [5–7,16,28], and were found to be in fair agreement with these references and Demirkaya et al. [11] and Goswami [12]. In addition, the PV solar panels efficiency values reported elsewhere in this paper were also found to be in agreement with what has been published, namely, Faragli [8], and Sami [24–26]. Moreover, it is worth mentioning that we found no data available in the literature for comparison purposes on the effciency of the hybrid sytem composed of the PV-Thermal, ORC with cooling capabilities as presented in this paper.

## 6. Conclusions

In this study, the performance of nanofluids in a hybrid system composed of a PV-thermal driven ORC with cooling capabilities is presented. This study was intended to investigate the performance and characteristics of the hybrid system using nanofluids $Al_2O_3$, CuO, $Fe_3O_4$ and $SiO_2$ under different conditions: solar radiations, heat transfer fluid temperatures, nanofluid volumetric concentrations. In particular, the hybrid system efficiency and the cooling effect produced were the the main focuses of this study. It has been shown that the enhancement of the efficiency and the cooling effect produced by the hybrid system in question are significantly dependent on not only the solar radiation but also the nanofluids concentration and the type of nanofluid as well as the heat transfer fluid temperature driving the ORC. It was found that the higher the nanofluid concentrations, solar radiation, and heat transfer fluid temperature, the higher the cooling effect. It also has been demonstrated that the higher the heat transfer fluid and the higher the nanofluid concentrations, the higher the hybrid system efficiency and the higher the cooling effect produced.

Moreover, it was also shown that on average, the hybrid system efficiency was higher by 17% with nanofluid CuO compared to water as the heat transfer fluid. In addition, it was also observed that the higher cooling effect produced is significantly increased with the use of the nanofluid CuO compared to the other nanofluids under investigation and water as the heat transfer fluid. It is believed that this is attributed to the fact that the nanofluid CuO has higher thermo-physical properties compared to the other nanofluids, including water, as the heat transfer fluid. This is an important finding since just by selecting the right nanofluid for a specific ORC design, the efficiency and cooling effect produced of this hybrid system can be significantly enhanced. It is also worth mentioning that the novelty of the

new concept presented in this paper is the hybrid system that includes PV-Thermal solar panels to drive an ORC cycle with the capability to produce a cooling effect. Finally, the results reported in this paper on ORC efficiency and PV solar panel efficiency were found to be comparable to what has been published in the literature.

**Supplementary Materials:** The following are available online at http://www.mdpi.com/2571-5577/3/1/12/s1, Table S1: Properties of Nanofluids.

**Funding:** This research received no external funding.

**Acknowledgments:** The research work presented in this paper was made possible through the support of the Catholic University of Cuenca.

**Conflicts of Interest:** The author declares no conflict of interest.

## Nomenclature

| | |
|---|---|
| $h_1$ | enthalpy at the outlet of the waste heat boiler (kj/Kg) |
| $h_2$ | enthalpy at the exit of the vapor turbine (kj/Kg) |
| $h_3$ | enthalpy at the condenser outlet (kj/kg) |
| $h_4$ | enthalpy at ORC pump outlet (kj/kg) |
| $h_5$ | enthalpy at inlet of cooling/freezing coil (kj/kg) |
| $h_6$ | enthalpy at outlet of cooling/freezing coil (kj/kg) |
| $h_7$ | enthalpy at the outlet of regenerator (kj/Kg) |
| $m_{ref}$ | refrigerant mass flow rate (kg/s) |
| $\alpha_{abs}$ | Overall absorption coefficient |
| G | Total Solar radiation incident on the PV module |
| $S_p$ | Total area of the PV module |
| $W_{ORC}$ | power produced by ORC (KW) |
| $p(t)$ | PV solar output (kW) defined by Equation (4) |
| $Qcc$ | cooling coil thermal capacity (kw) and defined by Equation (9) |
| $W_{P_{ORC}}$ | pump power consumption (9) |
| $Q_{in}$ | solar radiation (kw) and defined by Equation (1) |

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
