# Peer review of "Analysis of Nanofluids Behavior in a PV-Thermal-Driven Organic Rankine Cycle with Cooling Capability"

_asi, doi:10.3390/asi3010012_

Round 1

Reviewer 1 Report

In this manuscript, the performance of nanofluids in the PV thermal driven ORC system is studied numerically. The effects of nanofluid property and solar radiation on the system performance are analyzed. However, I would say that the manuscript is not well organized. A lot of relevant information correlating to the cycling system is missing from the paper. Thus, it isn't straightforward for the reviewer to evaluate the reliability and value of the results listed in the manuscript. And I believe it is also impossible for the reader to following this research properly. Besides, too many figures are listed in the paper without in-depth discussion.

In summary, I would suggest to reject it.

Author Response

Dear Reviewer No.1

Thank you for your time and comments and suggestions. We have integrated most of your suggestions as well as that of other reviewers into the text of the revised paper and in particular on the relevant cycling information. We hope your opinion of the paper will change.

Thanks,

Reviewer 2 Report

The author has analytically investigated the effect of different nanofluids in PV thermal driven ORC with a cooling coil. While the objective of the research seems interesting, the manuscript should be extensively revised before further assessment.

1) Overall, the manuscript is poorly written and the results are poorly presented.

2) The obtained results need to be verified, at least in part, with previous published experimental or numerical data.

3) The introduction needs to be rewritten as almost all of the paragraphs are not connected to each other. The author needs to cohesively explain the background of the work, the overview of the previous works and finally highlights the novelty of his work.

4) The parameters used for the modeling, e.g., heat transfer rates (Q) and temperatures (T), need to be illustrated in Figure 1.

5) The properties of the employed nanofluids need to be listed in a separate table.

6) For the sake of comparison, the results of different nanofluids need to be presented together. (In my personal opinion, using Excel format is not professional for presenting the results in a journal paper)

7) The captions of the figure are too short. The authors need to add more explanations.

Author Response

Dear Reviewer No.2

We greatly appreciate your time to read the paper and the comments and suggestions provided. We have integrated your comments about the introduction, figures and included separate table for the properties of the properties of the nanofluids.

Thanks again.

Reviewer 3 Report

The author discussed the performance of nanofluids in the PV-thermal driven organic Rankine cycle. The outcomes are quite obvious. It is clear, efficiency depends on input conditions and materials. In addition, the paper require deep revision of the manuscript. The methodology in the paper is not clear. These are the main issues that authors must resolve:

Abstract

Line 15-18, The result are have not a novelty and not interesting for the reader, please give more details like which nanoparticles’ performance is better. What is the novelty in your method?

Line 19, ‘simulation and discussion’’ .. replace by ‘’ simulation’’

Introduction

There is not any literature review for nanoparticles. The introduction should consist of a literature review of three parts:1) PV-thermal, 2) Organic Rankine Cycle, 3) Nanofluids

Line 28-29, there is no coherent structure between the paragraphs.

Line 37, 40 and 70 delete the Organic Rankine cycle and replace by ‘’ORC’’

Line 38 deletes the dot after [10].

Line 45, ‘’ Different thermodynamic …’’ that is not clear!

Line 48, new sentence start need dot after [11].

Section 2

Line 89, REFPROP, explain what is it?

Fig1- The quality is not good by zoom in.

Line 101,106 and 117 write parameters in one line (i.e ‘’ Where, alpha, G, Sp are …..’’)

Line 102, Is it the same as thermal panel efficiency ( Etta) in Eq. 10 in the reference?

Equations 2, 3 and 4 are not clear in the references 16,17 and 18. Which equation match with referenced papers?

Line 118, IL replace by IL

 Line 129, -Numbering of equations is shifted

Line 130, please replace by ‘’where h is enthalpy’’. Ans show the number of inputs and outputs in the figure.

Equation 11 is the same as equation 18 in reference 6? What is different?

Section 3

This section is the same as reference ‘6’, please reference and delete it or revise it.

Secction 4,

The quality of Figure 2 is not good. Instead of ‘’ solve mass, energy equations …’’ please mentioned the number of the equations.

Section 5,

This section require deep revision of the manuscript. The figures are not good quality. Please use professional graph plotting software like kaleidagraph. Also, please pay attention to the details of each figure. For example, in Figures 3-6, the range of solar radiation should be between 400-1300 and the y-axis should be between 0.2-0.7. Also, it should better display multiple images in one figure and subplot by A)..B)… Also, please pay attention to the subscripts of nanoparticles like Sio2 and Fe3O4. The colors must be unique for each material in all the paper. For example, blue for the water, CUO is orange and so on.

Line 208, [18} through [20] replace by [18-20].

Line 227, the end of the sentence needs a dot.

Line 270-276, These results have not any novelty.

Line 285 ‘’Mpreover’’ replace by ‘’Moreover’’.

Line 287, Figure.19 through.21 replace by Figure19-21.

Line 289, ‘’presentedn’’ replace by ‘’presented’’

Conclusion

It is not clear from the conclusions what is the novelty of the paper.

Line 313, what do you mean about ‘’slightly higher’’? Please give the quantity or percentage.

References

The font size and themes are different. Also, the citation style is different such as references 19 and 20.

Author Response

Reviewer No 3

We are in debt to you for the time taken to read and comment on the paper line by line.  We also responded and integrated your comments about equations, and included in the conclusions a statement about the novelty of the paper. Please note that the shifted numbering of equation should be taken care of when the paper is published.

Thanks again for valuable time to review the paper.

Round 2

Reviewer 2 Report

The author has partially revised his manuscript and did not address all the comments.

Still, the paper's quality is low and cannot be accepted for publication.

Mainly, the manuscript is full of typos, the introduction does not provide the sufficient background information highlighting the scope of the paper and its novelties, it lacks in-depth discussions, and the results are not validated. 

Author Response

Dear Reviewer #2

Greatly appreciate your comments, however, please note the following;

The revised version, addressed all your 7 concerns mentioned in your first review. Sufficient background is provided and changes have been made to the introduction section to address your comments regarding the scope and novelty of the concept presented in the paper. We have addressed in the previous version the issue of validation and the lack of experimental data on the subject matter presented in the paper.

Reviewer 3 Report

Thanks, the quality of the paper is much better but still, major problem for the quality of images still remains.
Please increase the quality of Figure 2. ( Should be good to 300% zoom in)
Figures (3-15) the range in the X-axis should be between 400-1300.
Figures (3,4,5 and 6) should be in display multiple images in one figure and subplot by A)..B)… Also, please pay attention to the subscripts of nanoparticles like Sio2 and Fe3O4. The colors must be unique for each material in all the paper.
Figures (7,8,9 and 10) should be in display multiple images in one figure and subplot by A)..B)…
Figures (11,12,13 and 14) should be in display multiple images in one figure and subplot by A)..B)…
Still, problem in writing remains like line 380 and 383 : [5,7, 16, 28},
Fe304 and Sio2
After reference 14 it must be 15, not 24, 25 and …. Please check the format and style of references with Endnote.
The format and style of highlight citation are different from other references.

Author Response

Dear Reviewer #3

Greatly appreciate your valuable comments. Please note the following;

We improved on the quality of images. Increased the quality of Figure.2 The x-axis scale in Figures 3 through 15 are imposed by the plotting program Attention has been paid to the subscriits of nanofluids and were corrected in the paper. We feel that Figures 7 through 14 better presented separately. References [15,7,16,28] have been corrected. References have been arranged to be in the right order in the paper. Format and style of the references have been changed by the MPDI formation, and during editing process will be taken care of.

Round 3

Reviewer 2 Report

The quality of the manuscript has improved compared to the initial submission. If the editor finds it suitable for publication, I have the following minor comments:

1) The presented results are in Excel format which is not suitable for publication. Also, the range of chosen x and y axes should be corrected.

2) What is "AI2O3"? is it Aluminium oxide? (Al2O3)? Also, in most figures (e.g., Figures 6,10, 14), and Table 1 it is written as "Ai2O3"!

3) The font type and quality of Figure 2 is unacceptably low. The author needs to replace it with a higher quality one.

4) As indicated before, at least some of the defined parameters should be illustrated in Figure 1.